# Fine Mapping of Glutamate Decarboxylase 65 Epitopes Reveals Dependency on Hydrophobic Amino Acids for Specific Interactions

**DOI:** 10.3390/ijms20122909

**Published:** 2019-06-14

**Authors:** Niccolò Valdarnini, Bettina Holm, Paul Hansen, Paolo Rovero, Gunnar Houen, Nicole Trier

**Affiliations:** 1Interdepartmental Laboratory of Peptide and Protein Chemistry and Biology, Department of NeuroFarBa, University of Florence, Via Ugo Schiff, 6, 50019 Sesto Fiorentino, Italy; valdarnini91@gmail.com (N.V.); paolo.rovero@unifi.it (P.R.); 2Department of Clinical Immunology, Rigshospitalet, Ole Maaløes vej 26, 2200 Copenhagen N, Denmark; Bettina.eide.holm@regionh.dk; 3Department of Drug Design and Pharmacology, Universitetsparken 2, 2100 Copenhagen Ø, Denmark; prh@sund.ku.dk; 4Department of Neurology, Rigshospitalet Glostrup, Nordre Ringvej 57, 2600 Glostrup, Denmark; Gunnar.houen@regionh.dk

**Keywords:** antibody, antigen, glutamate decarboxylase, peptides, type 1 diabetes

## Abstract

Characterization of multiple antibody epitopes has revealed the necessity of specific groups of amino acid residues for reactivity. This applies to the majority of antibody–antigen interactions, where especially charged and hydrophilic amino acids have been reported to be essential for antibody reactivity. This study describes thorough characterization of glutamic acid decarboxylase (GAD) 65 antigenic epitopes, an immunodominant autoantigen in type 1 diabetes (T1D). As linear epitopes are sparsely described for GAD65 in T1D, we aimed to identify and thoroughly characterize two GAD65 antibodies using immunoassays. A monoclonal antibody recognized an epitope in the N-terminal domain of GAD65, ^8^FWSFGSE^14^, whereas a polyclonal antibody recognized two continuous epitopes in the C-terminal domain, corresponding to amino acids ^514^RTLED^518^ and ^549^PLGDKVNF^556^. Hydrophobic amino acids were essential for antibody reactivity, which was verified by competitive inhibition assays. Moreover, the epitopes were located in flexible linker regions and turn structures. These findings confirm the versatile nature of antibody–antigen interactions and describe potential continuous epitopes related to T1D, which predominantly have been proposed to be of discontinuous nature.

## 1. Introduction

The majority of epitopes currently analyzed are discontinuous [1]. It has been estimated that 70–90% of all epitopes are discontinuous. Characterization of discontinuous epitopes is rather complicated compared to continuous epitopes, also referred to as linear epitopes. Nevertheless, as the epitopes are composed of similar amino acids (aas), results obtained using continuous epitopes are directly related to discontinuous models.

The majority of epitopes are composed of 8–12 aas [1,2,3,4,5,6]. Except from antibodies recognizing modified epitopes, are epitopes seldom shorter than 6 aas, as antibody-antigen interactions depend on several aas for a stable interaction [7,8,9]. Secondary structures and structure flexibility are central elements for a stable antibody-antigen interaction. Thus are flexible regions, loops and turns, but also occasionally α-helices favored epitope structures [1,10,11,12,13]. Epitopes primarily located on the surface of proteins are often favored due to an increased antibody accessibility [1,4]. 

Epitope structures are often rich in hydrophilic and charged aas, especially Arg, Asp and Glu, and with a lower ratio of aliphatic hydrophobic aas, although several epitopes have been described that depend on hydrophobic aas for stable interactions [1,2,13,14,15,16]. Moreover, the aas Tyr and Trp have been found to be overrepresented in epitopes as well [1]. These findings have been supported by previous studies, claiming that Tyr, Trp and charged residues generally are preferred in protein–protein interfaces due to their capability to form a multitude of interactions [3,6,13,14,17,18,19,20]. Aas, such as the hydrophobic aas Pro and Gly, are often found in epitopes as well, as these aas often are represented in turns and flexible regions [1,6,13]. Besides individual aa preferences, specific aa pairs have been observed in epitopes, suggesting that some aa pairs, e.g., Tyr:Tyr, Cys:Pro, Asn:Tyr. Asp:Pro, Arg:Tyr, Asn:Tyr and His:Tyr, work cooperatively in mediating antibody binding [1].

The number of aas essential for antibody reactivity varies depending on the type of antibody–antigen interaction. For example, an interaction with a high affinity may experience a 1000-fold reduction in antibody reactivity when substituting an essential aa, whereas a neighbour aa may be substituted without interfering with the affinity, although they are a central part of an epitope structure [6,7,9,20]. This may apply to the fact that some aas contribute with a specific side chain contribution (e.g., ionic bond or hydrophobic interaction) for a stable antibody–antigen interaction, whereas other aas contribute with backbone interactions (hydrogen bonds), which in theory can be contributed by any aa. The substitution of a central aa, which contributes with backbone interactions does not alter the secondary structure of the epitope unless its side chain interferes with other interactions [8,9,13]. 

Several assays for epitope characterization exist, but one of the most straightforward methods is to use immunoassays [5,7,10,13,19]. In these assays, usually a protein or peptide is coupled or coated onto a solid surface, which is recognized by a primary antibody. Bound antibodies are usually determined using labelled secondary antibodies. This approach has been used with great success for characterization of several linear epitopes [2,3,10,13,19]. When using methods such as X-ray crystallography, significant amounts of data are obtained at the atomic level in relation to specific interactions between the individual aas of the antibody and antigen, nevertheless, this type of epitope characterization is rather time consuming and requires a notable amount of material for the generation of suitable crystals [21,22]. 

Knowledge of epitopes related to the glutamic acid decarboxylase (GAD) protein is sparse. GAD is a pyridoxal phosphate (PLP)-dependent enzyme that catalyzes the conversion of L-glutamic acid to the inhibitory neurotransmitter γ-aminobutyric acid (GABA) and is among others expressed in the pancreatic β-cells [23,24,25]. In mammals, two isoforms of GAD exist, GAD67 and GAD65, based on their molecular weight. The isoforms show a general overall sequence similarity, with the middle and C-terminal domains having 74% identity, but differing (25% identity) in the N-terminal domain, mainly in the first 100 aas [23,24]. The two isoforms differ in their enzyme activity. GAD67 is located in the cytoplasm, is constitutively active, and provides for the basal production of GABA, whereas GAD65 is present in synaptic vesicles (besides the pancreatic β-cells) and undergoes auto-inactivation during enzyme activity, which occurs in the cell providing for a pulse in production under circumstances that demand a rapid surge of GABA synthesis and release [23,24,25,26].

The two isoforms are divided into three functional domains, an amino-terminal domain, a middle domain where the catalytic site resides (PLP-binding domain), and a carboxy-terminal domain. Roughly, antibodies to GAD67 are primarily associated with stiff-person syndrome (SPS) and autoimmune polyendocrine syndrome type 1, whereas GAD65 is a dominant autoantigen in type 1 diabetes (T1D) [2,24,25,26,27,28,29]. 

Epitope specificity in GAD-related neurological diseases has been examined only for SPS, indicating that the antibodies are directed against continuous GAD epitopes in all 3 domains, but predominantly against the catalytic region [30]. In contrast, GAD epitopes related to T1D are often directed to discontinuous GAD epitopes located in the middle and C-terminal domains [31]. Most T1D sera contain two distinct GAD antibody specificities, one of which targets an epitope region in the middle-third of GAD65 (aas 221–359) and one of which targets the carboxy-third of GAD65 (aas 453–569) [31,32]. Only few epitopes in the N-terminal region have been described. 

As epitope characterization of discontinuous epitopes can be rather challenging, no in depth characterization of GAD65 epitopes has been conducted until now. A single study describes reactivity to continuous GAD65 epitopes in patients with slowly progressive T1D. Here, GAD65 antibodies reacted specifically with an N-terminal linear epitope in the membrane-anchoring domain between aas 17–51 [33]. Furthermore, a conformational epitope between aas 244–360 was identified in a group of young individuals with acute onset T1D [33]. However, no study currently has described specific aa contributions to the antigen–antibody interaction between GAD65 and respective autoantibodies. 

In this study, three continuous GAD65 epitopes were thoroughly characterized with emphasis on essential aas necessary for antibody recognition and aa side chain functionality in order to obtain further knowledge of the specific interactions between GAD65 and autoantibodies related to T1D. 

## 2. Results

### 2.1. Screening of Overlapping Peptides Identifying Antigenic Regions 

To identify potential epitopes of two GAD65 antibodies, screening of overlapping resin-bound peptides covering the respective immunogens (aa 4–22 for the mAb GAD65 and aa 353–585 for the pAb GAD65) was conducted by modified enzyme-linked immunosorbent assay (ELISA). 

Figure 1a illustrates the reactivity of the mAb to peptides originating from the N-terminus of GAD65. Significant antibody reactivity was found to peptide 1 (aa 1–20, MASPGSGFWSFGSEDGSGDS). No antibody reactivity was found to peptide 2 (aa 10–30, FGSEDGSGDSENPGTARAWC) or 3 (aa 20–40, ENPGTARAWCQVAQKFTGGI). These findings indicate that the epitope of the mAb is located in the first 10 aas of the peptide or within the common overlap of peptides 1 and 2 (approximately aas 5–15). 

Figure 1b illustrates the reactivity of the pAb to overlapping peptides covering the aas 353–585. Distinct reactivity was found to peptides 51 (aa 501–520, HTNVCFWYIPPSLRTLEDNE) and 52 (aa 511–530, PSLRTLEDNEERMSRLSKVA), indicating that the epitope is found in the aas 511–520, PSLRTLEDNE. Reactivity was observed to peptides 38, 41, 49, 55 and 57 as well. However, antibody reactivity to peptides 38, 41, 49 and 57 was ascribed to nonspecific reactivity, as a control antibody of irrelevant specificity, reacted with these peptides as well (Figure 1c). Based on these findings, peptide 52 (PSLRTLEDNEERMSRLSKVA) and 55 (GTTMVSYQPLGDKVNFFRMV) were selected for further analyses.

### 2.2. Screening of N-Terminally and C-Terminally Truncated Peptides 

To identify the terminal epitope borders of the GAD antibodies, N- and C-terminal truncated resin-bound peptides were screened for antibody reactivity by modified ELISA. Peptides 1 (MASPGSGFWSFGSEDGSGDS), 52 (PSLRTLEDNEERMSRLSKVA) and 55 (GTTMVSYQPLGDKVNFFRMV) were used as templates for the generation of truncated peptides. 

Figure 2 represents GAD65 antibody reactivity to the terminally truncated peptides. For the mAb, the N-terminal aa Phe (aa 8) and the C-terminal aa Glu (aa 14) were essential for antibody reactivity, which is illustrated by significant reactivity to the N-terminal truncated peptide FWSFGSEDGGG and the C-terminal truncated GSGFWSFGSE (Figure 2a,b). No reactivity was found to the further truncated peptides (WSFGSEDGGG and GSGFWSFGS). Based on these findings, the minimum functional epitope of the mAb was identified as the aa sequence ^8^FWSFGSE^14^. 

For the pAb, which recognized two continuous epitopes, notable antibody reactivity was found to the N-terminal truncated peptide TLEDNEERMSRLSKV and the C-terminal truncated peptide PSLRTLED (Figure 2c,d), although the antibody reactivity to the RTLEDNEERMSRLSKV containing an N-terminal Arg was higher. Only truncated peptides containing the aas TLED reacted with the pAb, indicating that the TLED motif constitutes a central part of the epitope. 

Screening of GAD65 pAb reactivity to truncated analogues of peptide 55 (GTTMVSYQPLGDKVNFFRMV), illustrated that the N-terminal truncated peptide LGDKVNFF and the C-terminal truncated peptide MVSYQPLGDKV were the shortest peptides to be recognized (Figure 2e,f) although the peptide with a N-terminal Pro obtained a higher reactivity compared to the LGDKVNFF peptide. Further antibody reactivity to the peptides MVSYQPLGDKV and MVSYQPLGDKVN was low. Peptides with further truncated residues were not recognized by the pAb. Based on these findings, the aas to react significantly with the pAb were identified as ^515^TLED^518^ and ^550^LGDKV^554^. 

### 2.3. Final Identification of Complete Epitopes

To determine the complete GAD65 epitopes, competitive inhibition assays were conducted using relevant truncated peptide analogues.

Figure 3a depicts the reactivity of the mAb to the truncated peptides FWSFGSE-GSGFWSFGSEDGSG. As seen, all of the truncated peptides inhibited reactivity to the control peptide (PGSGFWSFGSEDGSGDSEN), confirming that the aa sequence ^8^FWSFGSE^14^ constitutes the complete epitope of the mAb. 

When analyzing antibody reactivity of the pAb to the truncated peptides (TLED-SLRTLEDNEE), containing the central TLED motif, it was observed that the four aas alone were not sufficient to inhibit antibody reactivity to the control peptide (peptide 52) (Figure 3b). The peptide RTLED was the shortest peptide, which inhibited antibody reactivity to the same level as the 10mer peptide (SLTREDNEE), indicating that the aas ^514^RTLED^518^ constitutes the complete epitope 1 of the pAb. 

Finally, the second continuous epitope of the pAb was determined using truncated peptides (PLGDKV-YQPLGDKVNFFR) (Figure 3c). As seen, the peptide PLGDKVN inhibited antibody reactivity by approximately 50%, whereas the remaining peptides PLGDKVNF-YQPLGDKVNFFR inhibited antibody reactivity similar to the control peptide (peptide 55), indicating that the aas ^549^PLGDKVNF^556^ constitute the second complete continuous epitope of the pAb. 

### 2.4. Identification of Essential Amino Acids 

To determine the contribution of each aa in the identified epitopes to the specific antibody interaction, competitive inhibition assays were conducted using Ala-substituted peptides. 

Figure 4a illustrates the reactivity of the mAb to Ala-substituted peptides, using the peptide FWSFGSE as template. As seen, peptides containing Ala in position 1 and 2 did not inhibit antibody reactivity, illustrating that the aas in these positions (Phe, Trp) were essential for reactivity. Ser in position 6 was not essential for antibody reactivity, as the peptide containing Ala in this position inhibited antibody reactivity to the same level as the control. Antibody reactivity was inhibited by approximately 60% when aas in position 3, 4, 5 and 7 were substituted with Ala. 

Figure 4b illustrates the reactivity of the pAb to Ala analogues of the RTLED peptide. As seen, Asp in position 5 and to some extent Thr and Leu in position 2 and 3, respectively, were essential for antibody reactivity, as peptides containing Ala in these positions did not inhibit antibody reactivity compared to the controls (RTLED and peptide 52). In contrast, the aas Arg and Glu were not essential for antibody reactivity, as substitution of these did not influence antibody reactivity. 

Figure 4c illustrates the reactivity of the pAb to Ala-substituted peptides using the peptide PLGDKVNF as template. As seen, primarily the aa Leu in position 2 was essential for antibody reactivity, as the peptide containing an Ala residue in this position did not inhibit antibody reactivity. The motif GDK was to some degree essential for reactivity, as substitution of these aas, reduced antibody reactivity. The remaining Ala-substituted peptides inhibited antibody reactivity to some degree compared to the control (PLGDKVNF), indicating that they were not essential for a direct contact through their side chains with the pAb [2]. 

Finally, functionality substituted peptides were screened for antibody reactivity, to determine whether the specific aa contribution to the antibody–antigen interaction relates to side chain functionality or the specific side chain. Various aa-substituted analogues were tested for antibody reactivity in competitive immunoassays. 

For the mAb, the aas in position 1 and 4 were essential for reactivity, as peptides containing aas of similar functionality in these positions did not inhibit antibody reactivity. Only Ser in position 3 and 6 could be substituted with Thr without influencing antibody reactivity (Figure 4d). Peptides containing substituted aas in the remaining positions inhibited antibody reactivity by approximately 60%.

Analysis of the antibody reactivity of the pAb to the RLTED epitope, revealed that only Asp in position 5 could not be replaced with an aa of similar functionality without interfering with antibody reactivity [Figure 4e]. The remaining aas in positions 1–4 were replaced with aas with similar functionality and maintained the ability of the peptide to inhibit antibody reactivity. The same applies to the PLGDKVNF epitope. Antibody reactivity was partly reduced when replacing Leu with Ile, the remaining aas were replaced with aas of similar functionality without interfering with antibody recognition (Figure 4f). 

### 2.5. Epitope Structures

To identify the orientation of the identified epitopes, the crystal structure of GAD65 was examined using RasWin software. Unfortunately, the N-terminal structure of GAD has not been crystalized, thus no structure could be assigned to the mAb epitope. However, given that the epitope is located in the first 20 aas of the proteins, the epitope is most likely localized in a flexible region without any ordered structure, which applies to the majority of terminal epitopes [1,34].

The epitopes of the pAb were verified by the crystal structure. Epitope ^514^RTLED^518^ (colored in red) was found in a flexible region connecting an α-helix (green) and a β-strand (blue). Unfortunately, the structure of aas 518–519 was not verified in the crystal structure. These aas are represented by red (aa 518) and grey (aa 519) dots. 2 aas at this position are not likely to alter the complete protein structure. 

Epitope 2 (^549^PLGDKVNF^556^) of the pAb is located in a flexible turn connecting two strands. As seen, the essential Leu residues in the epitopes are accessible (Figure 5c,d). 

## 3. Discussion

The present study describes epitope characterization of antibodies to GAD65. The mAb recognized a single epitope, whereas the pAb recognized two linear epitopes. It remains to be determined whether the pAb recognized two linear epitopes due to the structural nature of the immunogen applied, as it remains unknown, whether a native or non-native immunogen was used for immunization. The mAb, originally generated to aas 4–22, recognized the aa sequence ^8^FWSFGSE^14^ (Figure 2 and Figure 3). Especially the aas Phe^8^ and Trp^9^ were essential for antibody reactivity, as peptides containing Ala in these positions did not inhibit antibody reactivity and as the antibody did not recognize peptide 2 (FGSEDGSGDSENPGTARAWC), lacking these two aas. The specific side chain of Phe^8^ was essential to the binding of the antibody, as it could not be replaced with Ala or an aa of similar functionality. In contrast, Trp could not be replaced with Ala but to some degree with Phe, indicating that a large hydrophobic aa in this position is sufficient to sustain antibody reactivity. The remaining aas could to some degree be substituted with Ala and still inhibit antibody reactivity. 

The pAb originally directed to aas 353–585, recognized 2 epitopes, the aas ^514^RTLED^518^ and ^549^PLGDKVNF^556^. For the RLTED epitope, the aas Thr and Leu inhibited antibody reactivity by approximately 50% when inhibited with Ala, but not when substituted with aas of similar functionality, indicating that these aas contribute to sustain a stable structure. Alternatively, may Leu contribute with hydrophobic interactions, as replacement of Ala to some degree inhibited antibody reactivity, whereas replacement with Ile did not influence antibody reactivity. This remains to be determined. Only Asp appeared to interact directly through the side chain with the antibody, as substitution with Ala reduced antibody reactivity. Asp could, however, to some degree be replaced with Glu, indicating that a negatively charged aa in theory is sufficient to obtain antibody reactivity, whereas the remaining aas most likely contribute to maintain a stable conformation. A similar reactivity pattern was observed for the PLGDKVNF epitope. The majority of the aas could be replaced with Ala or an aa of similar functionality without interfering with antibody reactivity. Only Leu could not be replaced with Ala without reducing antibody reactivity, whereas Leu could be replaced with Ile without interfering with antibody reactivity. 

As seen for both the mAb and the pAb, peptide backbone and specific aa interactions were essential for antibody binding. These findings are in accordance to previous studies describing that some epitopes have a significant dependency of specific aa side chains in the epitope, whereas other epitopes only make direct contact through a single or two aa side chains [2,3,6,10,13,19]. The mAb appeared to be more dependent on specific aa contributions than the pAb. Aas that do not interact directly with the antibodies, although side chain interactions, may contribute to maintain a stable peptide structure, alternatively by contributing with hydrogen bonds though backbone interactions, this remains to be verified, preferably through x-ray crystallography. However, this is time consuming and requires large amounts of peptides and antibody.

The fact that the mAb seemingly was more dependent on specific aas for antibody recognition, may be explained by the nature of the antibodies. The mAb is more specific for a single epitope, whereas the pAb is less specific for the individual epitope, due to heterogeneity. The pAb represents a collection of antibodies that are directed to the complete fragment, aa 353–585. As a consequence, the pAb has a high overall affinity due to the recognition of multiple epitopes. When analyzing antibody reactivity to the two individual epitopes, the pAb was very tolerant to changes in the primary structure of the epitopes. One explanation to this observation is that the affinity of the pAb to the two epitopes is reduced, which was supported by prolonged developing times and reduced reactivity and ultimately increased tolerance to aa changes. This remains to be verified, e.g., by SPR experiments. 

As presented in this study, hydrophobic aas were essential for antibody reactivity. Originally, it was emphasized that hydrophilic amino acids are important for antibody–antigen interactions, which conform to that many epitopes are surface-oriented [1,4,14]. Several studies describe that epitopes are enriched in charged and polar aas and depleted of aliphatic hydrophic aas [2,3,4,10,13,16,35,36,37]. Especially the hydrophobic/constrained aa Pro has been reported to be found in epitope structures, as Pro often is represented in turn structures [13,15]. These findings are in accordance with computational characterization of B-cell epitopes conducted by Kringelum and colleagues, which indicate that epitopes often contain one or few hydrophobic aas in the center in combination with charged aas on the edge, with hydrophobic aas close to the antibody, positively charged aas further away and finally hydrophilic aas between the hydrophobic and charged aas [16]. Collectively, several interactions are in play to generate a stable antibody–antigen interaction. Based on the respective approximate binding energies for ionic interactions (–5 kcal/mol), hydrogen bonds (–4 kcal/mol) and hydrophobic interactions (–0.03) [38], ionic interactions and hydrogen bonds appear to be the main contributors for a stable antibody–antigen complex [2,3,6,10,13,19].

Structural analysis of the epitopes of the pAb revealed that the epitopes are located in flexible structures, which is in accordance to the literature [1,10,13]. The secondary structure of epitopes has been analyzed carefully, and has been reported to have a significantly lower portion of helices and strands and more loops and flexible regions compared to the composition of secondary structural elements of the entire antigen [1,14,39,40].

Antibody reactivity to the GAD67 isoform, with a sequence identity of approximately 74%, was determined as well, but the GAD65 antibodies were specific for GAD65 (results not shown), as the epitopes identified were not present in the GAD67 isoform. A similar study has been conducted using the GAD67 isoform as well [2]. Here, two epitopes were identified in the N-terminal of GAD67, to aa 14–23 and 91–99. Fine mapping of these revealed that antibody reactivity was related to amino acid side-chain functionality in combination with non-contact aas, which most likely contribute to a stable protein structure [2]. These findings conform to results described in this study, indicating that antibodies to the two GAD isoforms may experience similar reactivity patterns. For the epitope in the aa sequence 14–23, a hydrophilic Thr in combination with a charged Asp were essential for antibody reactivity, whereas for the epitope composed of the aas 91–99, hydrophobic aas were central for antibody reactivity [2]. These findings are in accordance to findings in this study, describing that hydrophobic interactions are essential for antibody reactivity. Collectively, these GAD antibodies appear to depend on a combination of specific interactions, rather than restricted to a single interaction. 

Antibody reactivity to GAD65 epitopes has been very sparsely described, which only identify potential antigenic regions rather than concrete epitope structures. This most likely relates to that the majority of the immunodominant epitopes of GAD65 are discontinuous [1,16]. This study has described antibody reactivity to continuous epitopes, which may aid in the process of identifying and characterizing potential GAD65 epitopes in T1D.

## 4. Materials and Methods 

### 4.1. Materials

Tentagel S NH_2_ resin was purchased from RAPP Polymere GmbH (Tübingen, Germany). Alkaline phosphatase (AP)-conjugated anti-mouse IgG and *para*-nitrophenylphosphate (*p*NPP) were purchased from Sigma Aldrich (Steinheim, Germany). Tris-Tween-NaCl (TTN) buffer (0.05 M Tris, 0.3 M NaCl, 1% Tween 20, pH 7.4), AP substrate buffer (1M diethanolamine, 0.5 mM MgCl_2_, pH 9.8), carbonate buffer (15 mM Na_2_CO_3_, 35 mM NaHCO_3_, 0.001% phenolred, pH 9.6) were from Statens Serum Institut (Hillerød, Denmark). Mouse GAD65 (clone no. 144) mAb was from GeneTex (immunogen: aa 4–22 of human GAD65) and rabbit GAD65 (clone no. 2) pAb was from Origene (immunogen: aa 353–585 of human GAD65). Overlapping peptides used for initial scanning and free peptides used for final epitope identification were purchased from Schäfer-N (Lyngby, Denmark). 

### 4.2. Peptides

The human GAD65 published sequence (Swiss-Prot: Q05329.1) was used to generate overlapping 20mer peptides, which covered the complete protein. Each peptide contained an overlap of 10 aas to the next peptide. In total, 58 peptides were generated by 9-fluorenylmetyloxycarbonyl solid-phase peptide synthesis. Peptides 1–3 were screened for reactivity to the mAb and peptides 36–58 were screened for reactivity to the pAb. Peptide 2 (aa 11–30), 55 (aa 511–530) and 57 (aa 510–560) were used as templates for generation of N-terminal and C-terminal truncated peptides. Ala- and functionality-substituted peptides were generated using the aa sequences ^8^FWSFGSE^14^, ^514^RTLED^518^ and ^549^PLGDKVNF^556^ as templates. Each aa was systematically substituted with Ala or an aa of similar functionality. All peptides are listed in Appendix A. 

### 4.3. Modified Enzyme-Linked Immunosorbent Assay

Resin-bound peptides were screened for reactivity as described elsewhere [7,10]. Briefly, resin-bound peptides (100 µL) (1 mg/mL), diluted 1:30 in TTN, were added to a 96-well multiscreen filterplate (Millipore, Copenhagen, Denmark) and rinsed with TTN buffer (250 µL/well) using a multiscreen vacuum manifold. PAb (1:1000) and mAb (1 µg/mL) (100 µL) were diluted in TTN added to the wells and incubated for 1 h at room temperature (RT) on a shaking table. Next, resins were rinsed 3 × 1 min as previously described and AP-conjugated anti-mouse IgG and anti-rabbit IgG (100 µL, 1 µg/mL) were added to the wells and incubated for 1 h at RT on a shaking table. Bound antibodies were quantified using *p*NPP (1 mg/mL, 100 µL) diluted in AP substrate buffer. Finally, the buffer was transferred to a Maxisorp microtitre plate (Nunc, Roskilde, Denmark) and the absorbance was measured at 405 nm, with background subtraction at 650 nm, on a Thermomax microtitre plate reader (Molecular Devices, Menlo Park, CA, USA).

### 4.4. Competitive Inhibition Assay

Maxisorp microtitre plates were coated with peptides 1, 52 and 55 (1 μg/mL) in carbonate buffer for 2 h at RT. In another plate, peptide analogues (1 mg/mL) and GAD65 antibody (mAb 1 µg/mL, pAb, 1:1000 dilution) were incubated for 1 h at RT. Next, the peptide-antibody samples were added to the coated microtitre wells and incubated for 1 h at RT on a shaking table, whereafter the wells were rinsed with TTN. Following this, the wells were incubated with AP-conjugated anti-mouse IgG or anti-rabbit IgG (100 µL, 1 µg/mL) for 1 h at RT on a shaking table. Bound antibodies were quantified using *p*NPP (1 mg/mL, 100 µL) diluted in AP substrate buffer and the absorbance was measured at 405 nm, with background subtraction at 650 nm, on a Thermomax microtitre plate reader (Molecular Devices, Menlo Park, CA, USA). 

### 4.5. Structural Analysis of GAD65 Epitopes

The software Raswin Cn3D 4.1 was used to identify the location and secondary structure of the identified mAb and pAb epitopes in human GAD65 (PDB ID: 2OKK).

## Figures and Tables

**Figure 1 ijms-20-02909-f001:**
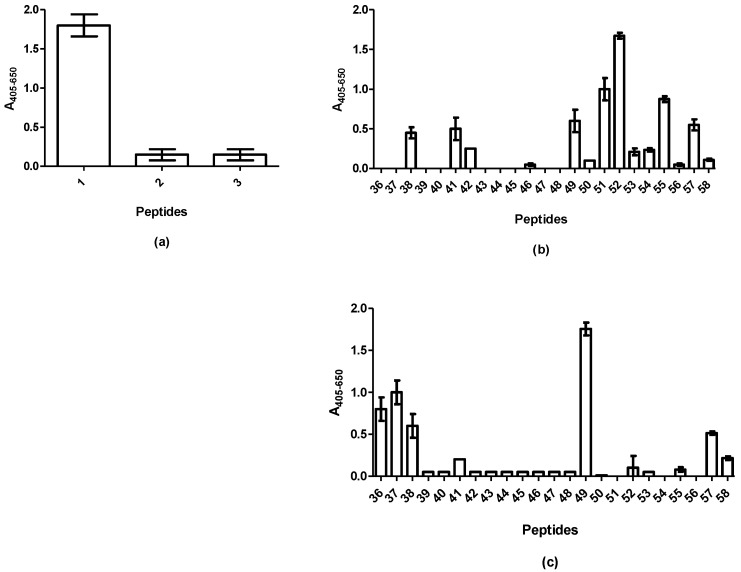
Reactivity of antibodies to GAD65 peptides analysed by modified enzyme-linked immunosorbent assay using resin-bound peptides. (**a**) Reactivity of the mAb GAD65 to peptides 1–3. Resin without peptides was used for background determination and subtracted. Peptide 1: aa 1–20, peptide 2: aa 10–30, peptide 3: aa 20–40. Peptide 3 was used as control peptide; (**b**) reactivity of the pAb GAD65 to overlapping 20mer peptides covering aas 353–585. Resin without peptides was used for background determination and subtracted; (**c**) reactivity of irrelevant antibody to overlapping peptides covering aas 353–585.

**Figure 2 ijms-20-02909-f002:**
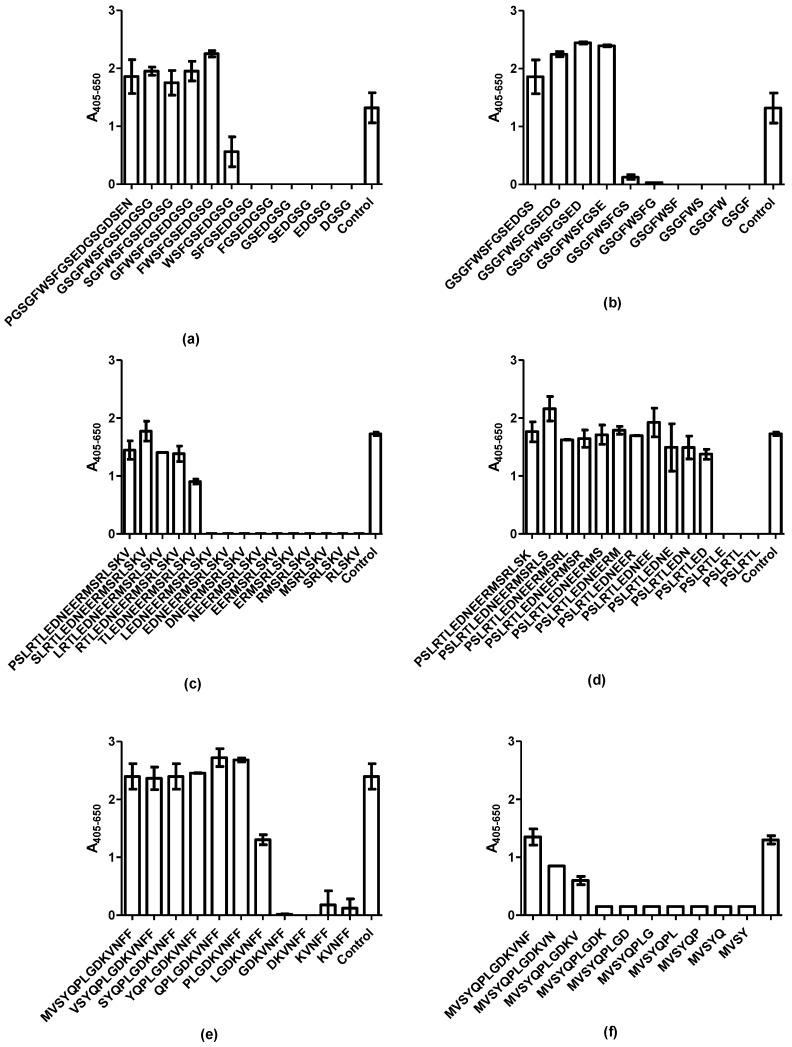
Reactivity of GAD65 antibodies to N- and C-terminally truncated peptides. (**a**) reactivity of the mAb GAD65 to N-terminally truncated peptides. Peptide 1 (MASPGSGFWSFGSEDGSGDS) was used as positive control; (**b**) reactivity of the mAb GAD65 to C-terminally truncated peptides. Peptide 1 was used as positive control; (**c**) reactivity of the pAb GAD65 to N-terminally truncated peptides of epitope 1. Peptide 52 (PSLRTLEDNEERMSRLSKVA) was used as positive control; (**d**) reactivity of the pAb GAD65 to C-terminally truncated peptides of epitope 1. Peptide 52 was used as positive control; (**e**) reactivity of the pAb GAD65 to N-terminally truncated peptides of epitope 2. Peptide 55 (GTTMVSYQPLGDKVNFFRMV) was used as positive control; (**f**) reactivity of the pAb GAD65 to C-terminally truncated peptides of epitope 2. Peptide 55 was used as positive control.

**Figure 3 ijms-20-02909-f003:**
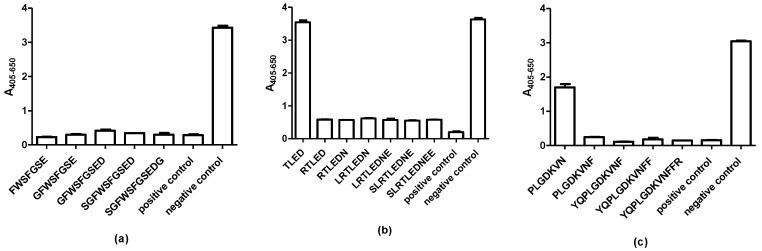
Verification of GAD65 antibody epitopes by competitive inhibition assay. (**a**) Reactivity of the mAb GAD65 to truncated peptides. The peptide PGSGFWSFGSEDGSGDSEN was used as a positive control, wells with no peptides added were used as negative control; (**b**) reactivity of the pAb GAD65 to epitope 1. Peptide 52 (PSLRTLEDNEERMSRLSKVA) was used as positive control, wells with no peptides added were used as negative control; (**c**) reactivity of the pAb GAD65 to epitope 2. Peptide 55 (TTMVSYQPLGDKVNFFRM) was used as positive control, wells with no peptides added were used as negative control.

**Figure 4 ijms-20-02909-f004:**
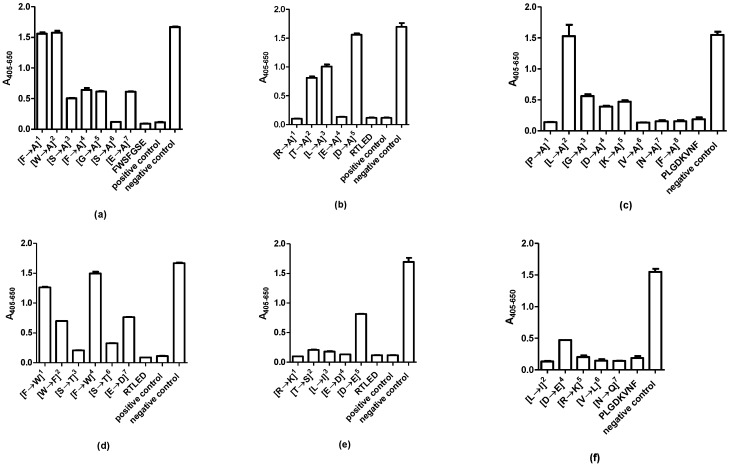
Reactivity of GAD65 antibodies to substituted peptide analogues. (**a**) Reactivity of the mAb GAD65 to Ala-substituted peptides. The peptides FWSFGSE and PGSGFWSFGSEDGSGDSEN were used as positive controls, wells without peptide inhibitors were used as negative control; (**b**) reactivity of the pAb GAD65 to Ala-substituted peptides. The peptide RTLED and peptide 52 (PSLRTLEDNEERMSRLSKVA) were used as positive controls, wells without peptide inhibitors were used as negative control; (**c**) reactivity of the pAb GAD65 to Ala-substituted peptides. The peptide PLGDKVNF were used as positive control, wells without peptide inhibitors were used as negative control; (**d**) reactivity of the mAb GAD65 to functionality-substituted peptides. The peptides FWSFGSE and PGSGFWSFGSEDGSGDSEN were used as positive controls, wells without peptide inhibitors were used as negative control. (**e**) reactivity of the pAb GAD65 to functionality-substituted peptides. The peptide RTLED and peptide 52 (PSLRTLEDNEERMSRLSKVA) were used as positive controls, wells without peptide inhibitors were used as negative control; (**f**) reactivity of the pAb GAD65 to functionality-substituted peptides. The peptide PLGDKVNF was used as positive control, wells without peptide inhibitors were used as negative control.

**Figure 5 ijms-20-02909-f005:**
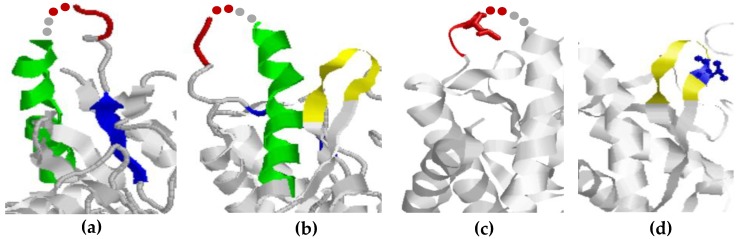
Epitope localization in the human GAD65 protein (PDB ID: 2OKK). (**a**) Epitope ^514^RTLED^518^ of the pAb GAD65 is colored in red. The structure of aas 518–519 was not verified in the crystal structure. These aas are represented by red (aa 518) and grey (aa 519) dots. The epitope is located in a flexible region connecting an α-helix (green) and a β-strand (blue); (**b**) The epitope ^549^PLGDKVNF^556^ of the pAb GAD65 is colored in yellow. The epitope is located in a turn connecting two strands; (**c**) Location of the potential important Leu (aa 516) residue in the first epitope of the pAb GAD65; (**d**) location of the potentially important Leu (aa 550) residue in the second epitope of the pAb GAD65.

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
