# Peer review of "Fine Mapping of Glutamate Decarboxylase 65 Epitopes Reveals Dependency on Hydrophobic Amino Acids for Specific Interactions"

_ijms, 2019, doi:10.3390/ijms20122909_

Round 1
Reviewer 1 Report
39: 70-90% of all epitopes are discontinuous. This might be true. Therefore, it is at least a little bit inconsequential to completely focus on continuous epitopes in this work.
53: "due to their capability to form a multitude of interactions". I do not agree with this notion. I think that "hydrophobic interactions", which are caused by water repulsion, is the main contribution.
63: Van der Waal bonds. Van der Waals bonds are very weak and do not lead to significant contributions in most cases.
69-71: without reading the references, the sentences are difficult to understand.
286: rexognized
347: "Only limited information is available, which describes antibody reactivity to discontinuous epitopes in the central domains. This study has described antibody reactivity to continuous epitopes, which may aid in the process of identifying potential GAD65 epitopes in T1D." I do not understand this statement.
How would it be possible to identify discontinuous epitopes?
The manuscript and the characterization of the continuous epitopes is interesting and fine. However, the introduction and discussion about interactions should be revised. In addition, the relevance of discontinuous epitopes for this paper is not clear to me.
Author Response
39: 70-90% of all epitopes are discontinuous. This might be true. Therefore, it is at least a little bit inconsequential to completely focus on continuous epitopes in this work.
In theory yes, but as mentioned in the introduction, characterization of antigen-antibody interactions using linear epitopes is also representative for discontinuous epitopes, as individual amino acid interactions are similar independent of epitope type. In the future, we want to focus on discontinuous epitopes, but this will require more substantial work.
53: "due to their capability to form a multitude of interactions". I do not agree with this notion. I think that "hydrophobic interactions", which are caused by water repulsion, is the main contribution.
We acknowledge your opinion. The sentence has been rephrased. However, several studies describe the importance of hydrophilic and charged amino acids as well.
63: Van der Waal bonds. Van der Waals bonds are very weak and do not lead to significant contributions in most cases.
We agree with this view. The contribution of Van der Waals interactions has been removed. However, although weak, many Van der Waals interactions may possibly play a minor role.
69-71: without reading the references, the sentences are difficult to understand.
The sentences have been rephrased and elaborated.
286: rexognized
Corrected.
347: "Only limited information is available, which describes antibody reactivity to discontinuous epitopes in the central domains. This study has described antibody reactivity to continuous epitopes, which may aid in the process of identifying potential GAD65 epitopes in T1D." I do not understand this statement.
The sentences have been rephrased.
How would it be possible to identify discontinuous epitopes?
Discontinuous epitopes may be identified by mass spectrometry in combination with chemical crosslinking. In the future, we want to focus on discontinuous epitopes using this approach, but this will require more substantial work.
The manuscript and the characterization of the continuous epitopes is interesting and fine. However, the introduction and discussion about interactions should be revised. In addition, the relevance of discontinuous epitopes for this paper is not clear to me.
Amended as requested. The introduction and discussion have been modified in relation to the importance of hydrophobic interactions. As mentioned previously, the amino acid interactions based on characterization of linear epitopes are representative for discontinuous epitopes, as amino acid interactions are similar independent of epitope type.
Reviewer 2 Report
The authors present precise delineation of the epitopes of a monoclonal and a polyclonal antibody directed against the immunogen GAD65, which is an autoimmunogen with clinical relevance. In the study, the relevant epitopes are identified and even the amino acids crucial for antibody binding are delineated. Identified peptides are modelled in the structure of GAD65. The study is well performed and precise, but describes only peptide mapping of antibody epitopes. To add value to the manuscript, the results could be substantiated by modelling of the identified relevant peptides. Results obtained could be compared with the report describing a similar study of a related enzyme. Materials and methods should be reported with more detail. I would sincerely recommend a re-read from the authors because certain sentences repeat throughout the article and not all sentences make sense. Please find below a list of suggested corrections.
Line 26: monoclonal antibody (no comma)
Line 23 and 29: thorough characterization repeats; not necessary in an abstract
Line 28: a polyclonal antibody (no comma)
Line 41: (aas)
Line 46: Thus are
Line 48: Epitopes primarily located on the surface of proteins are often favoured due to an increased antibody accessibility.
Line 50: and with a lower ratio of aliphatic
Line 63: van der Waals
Line 80: in the N-terminal
Line 82: it is either steady level production or basal production
Line 97: only few epitopes in the N-terminal region were described
Line 102: Further instead of in contrast: are the 2 studies you describe complementing or do they really report contrasting results?
Line 137: whole section 2.2: N-terminally and C-terminally truncated peptides
Line 156: I think this sequence should read LGDKVNFF. Please cite the sequence of the peptide 55 at the beginning of the paragraph.
Line 157: Peptides with further truncated residues
Figure 3, figure 4: legends to the figures should be self-explaining, so please specify which in the experiments is the positive and which the negative control
Line 253: the premise that the region is unstructured should be based on a reference or homology modelling or a closely related protein, the sentence is not acceptable as a general statement
Line 267: legend to figure 6: potentially important residue
Line280: please reword; 2 amino acids at this position are not likely to alter protein structure or similar
Line 285: What do you mean by most natural?
Line 286: recognized
Line 288: this statement is too general: what do you mean by the nature of the immunogen?
Line 296: some degree
Line 297: to the binding of the antibody
Line 320: this was mentioned at least 3-times in the article until now, please omit the sentence
Line 324: hydrophobic interactions are as well important in antigen-antibody interactions, please reword and introduce relevant literature
Line 327: throughout the paragraph, the authors refer to the affinity of the antibodies. Nevertheless, these should be described more precisely: with SPR data or at least a dose-response ELISA.
Line 330: heterogeneity
Line 341: why respectively? Figure 4b says only D was relevant and could be replaced with E.
Line 341: In contrast: what do you mean? Which antibodies were specific for hydrophobic amino acids?
Line 346: antibody reactivity
Line 348: literature missing
Line 352-362: instructions text should be deleted
Line 369: there is no reference 144
Line 383: similar properties; functionality as shown by the authors themselves is different
Line 391: again reference 144
Material and methods section: not concise enough. Please supply the information on volumes, sources of detection antibodies, times of incubation, etc.
Author Response
The authors present precise delineation of the epitopes of a monoclonal and a polyclonal antibody directed against the immunogen GAD65, which is an autoimmunogen with clinical relevance. In the study, the relevant epitopes are identified and even the amino acids crucial for antibody binding are delineated. Identified peptides are modelled in the structure of GAD65. The study is well performed and precise, but describes only peptide mapping of antibody epitopes. To add value to the manuscript, the results could be substantiated by modelling of the identified relevant peptides. Results obtained could be compared with the report describing a similar study of a related enzyme. Materials and methods should be reported with more detail. I would sincerely recommend a re-read from the authors because certain sentences repeat throughout the article and not all sentences make sense. Please find below a list of suggested corrections.
Dear reviewer, thank you for your comments, which have improved the manuscript and contributed to a more precise manuscript. We agree that modelling of identified relevant peptides in theory could be interesting, however, we believe the epitopes identified are too short to fold up into stable secondary structures. Based on this knowledge further information about the identified epitopes would not be obtained by modelling. The antibodies have been analyzed for reactivity to the GAD67 isoform, a similar enzyme, primarily expressed in the brain. However, the antibodies did not recognize the GAD67 isoform, as the antibodies were directed to GAD65-specific regions. We previously characterized 2 mAbs directed to the N-terminal region of GAD67. This study has been added to the discussion and compared to results obtained using the GAD65 isoform. Moreover, the material and methods section have been amended to describe this in detail.
Line 26: monoclonal antibody (no comma) corrected.
Line 23 and 29: thorough characterization repeats; not necessary in an abstract corrected.
Line 28: a polyclonal antibody (no comma) corrected.
Line 41: (aas) corrected.
Line 46: Thus are corrected.
Line 48: Epitopes primarily located on the surface of proteins are often favoured due to an increased antibody accessibility. Corrected.
Line 50: and with a lower ratio of aliphatic corrected.
Line 63: van der Waals corrected.
Line 80: in the N-terminal corrected.
Line 82: it is either steady level production or basal production corrected.
Line 97: only few epitopes in the N-terminal region were described corrected.
Line 102: Further instead of in contrast: are the 2 studies you describe complementing or do they really report contrasting results? Corrected.
Line 137: whole section 2.2: N-terminally and C-terminally truncated peptides corrected..
Line 156: I think this sequence should read LGDKVNFF. Please cite the sequence of the peptide 55 at the beginning of the paragraph. Corrected.
Line 157: Peptides with further truncated residues corrected.
Figure 3, figure 4: legends to the figures should be self-explaining, so please specify which in the experiments is the positive and which the negative control corrected.
Line 253: the premise that the region is unstructured should be based on a reference or homology modelling or a closely related protein, the sentence is not acceptable as a general statement corrected.
Line 267: legend to figure 6: potentially important residue corrected.
Line280: please reword; 2 amino acids at this position are not likely to alter protein structure or similar corrected.
Line 285: What do you mean by most natural? The sentence has been rephrased to avoid confusion.
Line 286: recognized corrected.
Line 288: this statement is too general: what do you mean by the nature of the immunogen? The sentence has been rephrased.
Line 296: some degree corrected.
Line 297: to the binding of the antibody corrected.
Line 320: this was mentioned at least 3-times in the article until now, please omit the sentence. Paragraphs mentioning side chain specificity and backbone interactions have been rewritten to one paragraph.
Line 324: hydrophobic interactions are as well important in antigen-antibody interactions, please reword and introduce relevant literature. Corrected. This paragraph as well as a paragraph in the introduction have been rephrased.
Line 327: throughout the paragraph, the authors refer to the affinity of the antibodies. Nevertheless, these should be described more precisely: with SPR data or at least a dose-response ELISA. We acknowledge that it would be appropriate and interesting to conduct affinity studies with the identified epitopes. However, it can be quite challenging to work with peptides of this size. Two problems are commonly encountered when using SPR for peptide adsorption studies: the need to account for "bulk-shift" effects and the influence of peptide-peptide interactions at the surface. Bulk-shift effects represent the contribution of the bulk solute concentration to the SPR response that occurs in addition to the response due to adsorption. Peptide-peptide interactions, which are assumed to be zero for Langmuir adsorption, can greatly skew the isotherm shape and result in erroneous calculated values of delta Go ads. In relation to a dose-response ELISA problems are encountered as well. Coating difficulties are prevailing when using peptides of this size, as a consequence, final experiments conducted were conducted as inhibition assays in this study. Moreover, the concentration of the pAb used in this study is unknown, which would yield a misleading result. We will aim for the suggested studies in the future, but as this work was primarily carried out during an exchange student visit, we will plan for future separate studies.
Line 330: heterogeneity Corrected.
Line 341: why respectively? Figure 4b says only D was relevant and could be replaced with E. Corrected
Line 341: In contrast: what do you mean? Which antibodies were specific for hydrophobic amino acids? Corrected. The sentence has been rephrased.
Line 346: antibody reactivity Corrected.
Line 348: literature missing. A relevant reference has been added.
Line 352-362: instructions text should be deleted Corrected.
Line 369: there is no reference 144. Corrected. The number refers to the clone number. The sentence has been rephrased.
Line 383: similar properties; functionality as shown by the authors themselves is different Corrected
Line 391: again reference 144 Corrected.
Material and methods section: not concise enough. Please supply the information on volumes, sources of detection antibodies, times of incubation, etc. Corrected. The materials and methods section has been rephrased.
Round 2
Reviewer 2 Report
Dear Authors,
your correction was thorough and prompt and the manuscript has indeed attained a higher level. Please be so kind to consider:
Line 149: The sequence for peptide 52 is still missing
Line 407: a significantly lower proportion of helices and strands and more loops and flexible regions compared to the composition of secondary structural elements of the entire antigen
The manuscript has improved and can be recommended for publication.
The manuscript has been revised to meet the comments raised by the reviewer.